# Prediction of Diagnosis-Related Groups for Appendectomy Patients Using C4.5 and Neural Network

**DOI:** 10.3390/healthcare11111598

**Published:** 2023-05-30

**Authors:** Yi-Cheng Chiang, Yin-Chia Hsieh, Long-Chuan Lu, Shu-Yi Ou

**Affiliations:** 1Department of Information Management, National Chung-Cheng University, Chia-Yi 621301, Taiwan; sofia@mis.ccu.edu.tw (Y.-C.C.); syou@vghks.gov.tw (S.-Y.O.); 2Taichung Tzu-Chi Hospital, The Buddhist Tzu Chi Medical Foundation, Taichung 427213, Taiwan; 3Department of Business Administration, National Chung-Cheng University, Chia-Yi 621301, Taiwan; bmalcl@ccu.edu.tw

**Keywords:** diagnosis-related groups, DRG, C4.5, back-propagation neural network, appendectomy

## Abstract

Due to the increasing cost of health insurance, for decades, many countries have endeavored to constrain the cost of insurance by utilizing a DRG payment system. In most cases, under the DRG payment system, hospitals cannot exactly know which DRG code inpatients are until they are discharged. This paper focuses on the prediction of what DRG code appendectomy patients will be classified with when they are admitted to hospital. We utilize two models (or classifiers) constructed using the C4.5 algorithm and back-propagation neural network (BPN). We conducted experiments with the data collected from two hospitals. The results show that the accuracies of these two classification models can be up to 97.84% and 98.70%, respectively. According to the predicted DRG code, hospitals can effectively arrange medical resources with certainty, then, in turn, improve the quality of the medical care patients receive.

## 1. Introduction

Due to the gradual increase in medical expenditures each year, insurance institutions in many countries, such as Germany, Canada, Japan, Taiwan, etc., have adopted a prospective payment system (PPS) to control the increasing expenditures [1]. In the past, the payment system for health insurance has worked on a fee-for-service policy, which may have resulted in an incentive to over-supply medical services or cause a supplier-induced demand for medical resources. To improve the defect of fee-for-service and to control the abnormal increment of medical expenditures, many countries devote their efforts to the revolution of the payment system. Compared to fee-for-service payment, the PPS has the potential to reduce the waste of medical resources by encouraging a change in terms of reimbursement. Among the PPS, the global budget system is a well-known macro-control measure, while case payment is a micro-control measure [2,3]. Case payment aims to reduce costs by imposing financial responsibility on the service providers by providing fixed costs for a certain set of diseases, while DRG (diagnosis-related group) payment using a fixed price has a broader definition compared to case payment. Since commencing the cost-containment property, the DRG payment system has changed the behaviors of doctors and hospitals due to the fact that they are forced to participate in the allocation of financial and medical resources to ensure high-quality medical care and to monitor the increase in medical expenditures.

However, the DRG code of patients may not be completely confirmed until they are discharged [4]. In other words, the hospital does not know how many available medical resources can be reimbursed during the hospitalization of inpatients. If there was an earlier and more accurate understanding of the likely DRGs of incoming patients, administrators would be able to make more informed decisions about staffing levels, equipment needs, and other resource allocation decisions.

Acute appendicitis, an inflammation of the appendix, is the most common cause of emergency abdominal surgery [5,6,7]. The major diagnostic categories (MDCs) of patients with appendectomy are classified as MDC6 (diseases of the digestive system) according to the TDRG of Taiwan [8]. As shown in Table 1, the operating room procedures of appendectomy patients are 47.01, 47.09, 47.2, or 47.99, respectively, coded in ICD10-CM. From the table, it can be seen that the presence of complications or comorbidities (CCs) in patients is an important discriminator of DRG code for the patients. Precisely, patients, if classified as DRG 164 and DRG 167, receive the same principal diagnosis and operating room procedures as patients classified as DRG 165 and 167, except that the formers have CCs. However, if we only use the above attributes to classify patients, we cannot accurately obtain their DRG code until the patients’ discharge. Identifying which DRG the patient should be classified as before they are discharged is important to control costs for hospitals. The literature [9,10,11,12] shows that physiological risk factors have an influence on the health conditions and episodes of illness of individuals. One’s personal health condition reflects physiological levels that can be obtained through medical examinations and screenings. The aim of this paper is to use classification methods to obtain DRG codes using the medical examination data, gender, age, etc., of patients with appendectomies (i.e., input attributes). Through doing this, which DRG code a patient with appendectomy can be determined before his/her discharge.

Until now, there has been little research aimed at predicting the DRG code of inpatients with appendectomy before they are discharged. The aim of this paper is to predict the DRG code of inpatients suffering from appendectomy as early as possible. Since the information of an inpatient, such as their ICD-10 and complication and comorbidity, cannot be precisely known until he/she is discharged, this paper uses other vital signs data, which can be obtained while an inpatient is admitted to hospital in order to predict their code. Two popular algorithms, a decision tree C4.5 and a back-propagation neural network (BPN), are used for the classification.

Classification is a process of learning that maps a data item into one of several predefined classes [13,14,15]. The objective of learning is to create a classification model (or classifier) for prediction [16,17]. Decision tree is a popular classification method; C4.5 is a program used for inducing classification rules in the form of decision trees from a set of given examples [16,17,18]. C4.5 uses information to help decide which attributes have the largest amount of information to induce classification rules in the form of decision trees [18]. The advantages of a decision tree constructed by C4.5 are that (1) it can generate understood rules and (2) it can handle continuous values and class variables.

Neural network is another frequently used method applied in the classification problem [19,20]. Nethe neuraletwork uses a large number of artificial neurons to simulate the ability of biological neural networks [19]. Artificial neurons, which receive information from the outside environment, or other artificial neurons simulate the functions of biological neurons. After a simple calculation, these artificial neurons output the result to the outside environment or other artificial neurons to represent complicated relationships between input and output. The most popular type of neural network is the back-propagation neural network (BPN), which uses the gradient steepest descent method to minimize the error function [14]. The elements of the back-propagation neural network contain the input, hidden, and output layers. The input layer consists of neurons standing for input attributes, the hidden layer represents the interaction of the input layer, and the output layer contains neurons standing for output attributes. The advantages of BPNs are that they can construct a complicated nonlinear network model and learn patterns precisely [21]; in addition, their recall speed is very fast. A BPN is suitable for pattern recognition, classification problem, expert system, noise filter, data reduction function synthesis, and so on.

Similar to the C4.5 algorithm, BPNs have also been widely applied in the medical field. For example, a BPN was developed to predict the length of stay of an inpatient [22]. Walczak [23] has developed a medical decision support system based on BPNs to predict the transfusion requirements of patients during three-time slices: one for the first two hours, another from the second hour to the sixth hour, and the third from the sixth hour to the twenty-fourth hour. The accuracy of the support system can be up to 91.4% on average for the three-time slices. Wu and Su [24] recognized a gait pattern in ankle arthrodesis for distinguishing differences between healthy and pathological gait. The recognition mechanism is developed upon BPNs, and the correctness of the classification can be up to 95.8%.

Some researchers use BPNs and other techniques for classification. For example, Sakka and Koutsouris [25] used data mining techniques to detect calcifications in mammograms, whereas Su et al. [26] used fine needle aspiration cytology data to check whether a breast tumor was malignant and used tongue diagnosis image data to check whether upper GI was a disorder using a Bayesian network (BNs), C4.5, and a BPN. The best performance among these three algorithms was the BPN, whose accuracy was 96.0% in diagnosing breast tumors and 91.6% in diagnosing upper GI disorders.

Utilizing classification techniques to predict the DRG code of an appendectomy inpatient while he/she is admitted to hospital is practicable. This paper utilizes two models constructed using the C4.5 algorithm and back-propagation neural network (BPN) for the classification. By conducting extensive experiments, this paper compares these two models in predicting the DRG code of appendectomy patients. When an earlier classification is obtained, hospitals that are in competition with each other can easily allocate their scarce healthcare resource to the desired patient by adapting the suitable clinical pathway for different DRG codes in order to reduce the waste of medical resources [27]. Finally, the quality of the treatment patients receive can be improved when resources are used effectively.

## 2. Materials and Methods

### 2.1. Data Processing

The literature [11,12] has shown that physiological risk factors influence the health conditions and episodes of illness of individuals. One’s personal health condition reflects physiological levels, which could be obtained using medical examinations and screenings. This paper collected the medical examination data, gender, and age of patients with appendectomies (i.e., input attributes) so that we could predict which DRG the patient is before his/her discharge. In addition, we also identified which physical and physiological factors potentially influence DRG classifications.

In order to construct the classification model for the patients with appendectomy, the paper used the historical data of patients with appendectomy (as the training data) from two different regional hospitals in Taiwan (denoted as hospitals A and B) with 313 and 125 patients, respectively. Eligible patients were those who had undergone excision of the appendix whilst they were in hospital. In addition, the paper selected the examination data of these patients from their first day of hospitalization.

We then selected the examination data required by the physicians for the diagnosis of appendectomy. Note that the sorts of vital sign data from the two hospitals are different; that is, hospital A may have some vital sign data in its patient records, but hospital B does not, and vice versa. The classification models were constructed by using the same algorithm for the two hospitals based on their respective data. The attributes of examination data will be used to construct the classification model in the paper.

These appendectomy patients were classified into four DRGs (from DRG 164 to DRG 167). The medical dataset has a large discrepancy in the number of classes in nature. For example, only 6 patients were classified as DRG 165, but 290 patients were classified as DRG 167. If the imbalanced data are fed into the model for training, the model tends to learn the majority class at the expense of the minority class. Such a skewed class distribution problem leads to prediction without discrimination. The collected data are imbalanced data in the way that some DRG classes are a minority while some are a majority. Oversampling and undersampling in model training are techniques used to adjust the class distribution of a dataset [28]. This paper adopted oversampling, i.e., argument (copy) the observations in the minority class so that the number of observations of each DRG class was nearly equal [29]. To be precise, we made copies of the minority class data in order to increase the amount of minority class data. This type of oversampling has less bias and has been commonly adopted in the literature. Table 2 and Table 3 show the original sampling and their augmented sampling of the DRGs of the patients with appendectomy in hospitals A and B, respectively.

### 2.2. Classification Model Construction

This paper utilized a BPN to construct the classification model (called the classifier), which learns by setting up a training dataset and compares the prediction class of a network for each sample with the actual class. For each sample, the weights were modified in order to minimize the mean squared error (MSE) between the prediction class and the actual class. When the MSE between the two classes becomes stable, i.e., no more significant decrease, the neural network can be deemed to have finished training. In terms of the number of hidden layers, BPNs usually have good convergence behavior with one- or two-hidden layers. According to previous experience, the general solutions can be a one-hidden layer, and the complex ones can be a two-hidden layer [30]. The one-hidden layer was utilized in this paper. In addition, for a larger number of units in the hidden layer, the slower the network coverages, the smaller the error we will obtain. In this paper, we used the default formula (shown in Equation (1)) as a way of counting the number of units in the hidden layer.
*Number of units of the hidden layer = (input units + output units)/2*(1)

The BPN uses the gradient descent method with an adjustment of the learning rate and momentum factor to minimize the error function. Learning rate refers to the rate at which errors modify the weights, and a momentum factor was used in this paper to allow a previous weight change to influence the next weight change in this cycle. Based on experience, optimal parameters (i.e., learning rate and momentum factor), which result in the best convergence, were from 0.1 to 1.0 and from 0.1 to 0.9, respectively [30]. In this paper, the values of the learning rate and momentum factor were set from 0.1 to 1.0 and from 0.1 to 0.9, respectively, with a uniformly increased interval of 0.1.

We divided appendectomy patients into four groups (i.e., from DRG 164 to DRG 167) as their classes. The procedure for constructing the classifiers is shown in Figure 1. In the first step, for appendectomy patients, all the prediction attributes of the DRG mentioned in Table 3 were fed into the BPN or C4.5 classifiers, then we adjusted the parameters of the BPN in order to find the best BPN classifier in terms of accuracy. In the second step, in order to construct classifiers, the paper adopts genetic search and best-first algorithms [31] to enhance accuracy through the method of finding out which attributes are more representative among all attributes. The genetic search method uses a simple genetic algorithm, while the best-first method uses the way of greedy hill climbing to search for the appropriate attributes [31]. In this way, we also adjusted the parameters of the BPN in order to find the best classifier. Then, we compared the accuracies of the classifiers constructed by C4.5 and the BPN with the selected attributes. In most cases, the selected attributes filtered by genetic search or best-first algorithms can improve the accuracy of the classification model [31]. In this paper, we identified which one was better than the other in order to establish a more accurate classifier.

In the last step, we select the classifiers with the highest accuracy constructed using C4.5 and the BPN with all or selected attributes, respectively. We combine these classifiers using the bagging method in order to further enhance the classification accuracy. Breiman [4] proposed a bagging method based on the concept of “bootstrap aggregating”. The bagging method aims to manipulate the training dataset by randomly replacing the original training dataset in order to generate several new training datasets, namely, C_1_, C_2_, …, C_N_. The replacement training datasets are known as bootstrap replicates of the training dataset. Suppose we classify the new data sample, M, then each classifier returns its class prediction to determine the final class (i.e., vote). These classifiers count the votes and assign the class with the most votes (seen in Figure 2). Note that the RIFT Study Group [32] also conducted a series of experiments to evaluate prediction models for appendectomy patients. They focused on the prediction of whether a patient with acute right iliac fossa (RIF) pain had an appendectomy. Their data may include eligible patients who did not undergo surgery. The aim of this previous paper was to evaluate whether the prediction models used were as reliable as the clinical decision support system, while our paper focuses on the prediction of the DRG code (i.e., classification) for patients who have undergone surgery. Therefore, the motivation and focuses of that paper and the present paper are different.

## 3. Results and Discussion

### 3.1. Model Evaluation Index and Classification Results with All Attributes

To demonstrate the feasibility of the prediction of DRG codes with examination data for patients with appendectomy when they are admitted to hospital, this paper constructed classifiers using C4.5 and BPN algorithms, associated some enhancing methods, then evaluated the accuracies of these models. Our first method used to evaluate the robustness of a classifier was to perform cross-validation. In detail, the accuracy of the proposed classifiers was tested using tenfold cross-validation, meaning the collected dataset was divided into ten subsets, in which one subset was used as the testing dataset, and nine out of ten subsets were used as the training datasets. We then adopted the evaluation index, i.e., the precision rate, which is the number of predictions that are true and values that are positive (i.e., true positive) divided by the total number of predictions whose values are positive, used as the main index to evaluate the performance of the classifiers. The accuracy rate was also used to evaluate the performance of the prediction. Accuracy is the number of correct predictions out of all of the predictions. In detail, it is defined as the number of true-positive and true-negative predictions divided by the sum of the number of true-positive, true-negative, false-positive, and false-negative predictions. Other indexes, such as the recall rate and specificity, were not adopted in this paper since their information cannot provide estimates of the reimbursement fee reimbursed for the hospitals’ reference.

For the first step shown in Figure 1, this paper constructed classifiers with all of the attributes. Then, this paper analyzed and compared the accuracies of the classifiers of C4.5 and BPN algorithms. In general, precision and recall rates were used to evaluate the performance of the classifier under binary or even multiple classes. The precision of a classifier is defined as the number of correctly predicted members as a class out of all the predicted members of that class. The recall rate is different from the precision rate as it counts the ratio of the number of correctly predicted members as a class against the number of actual members belonging to that class. The recall rate is important as a classifier, especially in the medical field.

There are 26 prediction attributes and 1156 examples in hospital A. Table 4a shows the confusion matrix of the C4.5 classifier for hospital A. From Table 4a, we compute the accuracy of model as 95.42% ((290 + 288 + 288 + 237)/1156), and the precision rates from DRG 164 to DRG 167 were 99.3% (=290/292), 92% (=288/313), 91.7% (=288/314), and 100% (=237/237), respectively. In addition to the accuracy and precision, we are also curious to decipher the recall rate of each DRG code, which is the ratio of the number of correctly predicted members as a DRG code against the number of actual members belonging to that DRG code. The recall rate rates from DRG 164 to DRG 167 are 100.0% (i.e., 290/290), 100.0% (i.e., 288/288), 100.0% (i.e., 288/288), and 81.7% (i.e., 237/290), respectively. Note that DRG 167 stands for appendectomy patients with a diagnosis of “laparoscopic appendectomy without complicated principal diagnosis without CC (comorbidity and complication)”, while the other DRGs are appendectomy patients with a complicated diagnosis. In other words, the patients classified as DRG 167 are patients who are not seriously ill. In most cases, the complexity of an appendectomy patient cannot be quickly diagnosed upon his/her admission to hospital; some patients may worsen or be fully diagnosed after undergoing an advanced examination. Thus, this phenomenon can explain why the precision of a classifier to identify the patient belonging to DRG 167 is highest, while the recall rate of the patients belonging to DRG 167 is the lowest compared to those of other DRG codes.

There are 33 prediction attributes and 387 examples in hospital B. Table 4b shows the confusion matrix of the C4.5 classifier for hospital B. We computed the accuracy of the model to be 88.11% ((98 + 87 + 95 +61)/387), and the precision rates from DRG 164 to DRG 167 were 98% (i.e., 98/100), 77.6% (i.e., 87/112), 87.9% (i.e., 95/108), and 91% (i.e., 61/67), respectively. In addition, the recall rates of each DRG code from DRG 164 to DRG 167 are 100.0% (i.e., 98/98), 91.6% (87/95), 100.0% (95/95), and 61.6% (61/99), respectively. Similar to the above discussion for hospital A, the complexity of an appendectomy patient cannot be quickly diagnosed at his/her admission as some patients may worsen or not be fully diagnosed until after they have undergone an advanced examination. This reason can account for the phenomenon that the precision of a classifier to identify the patient belonging to DRG 167 is high, while the recall rate of the patients belonging to DRG 167 is the lowest compared to those of other DRG codes.

For the BPN classifier, we set the learning rate from 0.1 to 1.0, with an increased interval of 0.1 for each run of the test, then we evaluated their accuracy under the values of the momentum factors from 0.1 to 0.9, respectively. Figure 3a shows the analysis diagram of the classifier constructed using the BPN classifier with all the attributes of hospital A. The result revealed that the prediction accuracy was stable in the case of most learning rates, though it worsens when the learning rate is over 0.8 under the values of momentum factors of 0.8 and 0.9. Figure 3b displays a zoomed-in picture of Figure 3a, where the accuracy is at its best (i.e., 98.70%), which occurred when the learning rates were set as 0.7 and 0.8 under the values of their momentum factors of 0.7 and 0.9, respectively.

As illustrated in Figure 3b, when the learning rates are 0.8 and 0.9 individually under the values of the momentum factors of both 0.7 for hospital A, the confusion matrix (shown in Table 5) shows that the accuracies of the BPN classifier are 98.7% and 98.7%, respectively. They both consisted of TP rates from DRG 164 to DRG 167 of 100%, 100%, 100%, and 94.8%, respectively.

Figure 4 shows the analysis diagram of the classifier constructed using the BPN classifier with all of the attributes of hospital B. The result revealed that the classification performance is not stable; it became worse while the learning rate was increasing under the increasing values of the momentum factors. Figure 4b is the zoomed-in picture of Figure 4a, which shows when the accuracy reaches its best (i.e., 96.12%), occurring when the learning rates were set as 02 and 0.8 under the values of their momentum factors of 0.4 and 0.7, respectively.

As an illustration of Figure 4b, when the learning rates are 0.2 and 0.8 individually under the values of momentum factors of 0.4 and 0.7 for hospital B, the confusion matrix (shown in Table 6) shows that the accuracies of the BPN classifier are 96.12% and 96.12%, respectively, which consist of TP rates from DRG 164 to DRG 167 which are 100%, 100%, 100%, and 84.8%, respectively. From the above results for classifiers C4.5 and BPN for hospitals A or B, the accuracy of the classifiers constructed using BPN algorithms with all attributes is better than those constructed using C4.5 classifiers with all attributes. This result is the same as that obtained by [33], which indicated that the accuracy of the classifier trained by a neuron network was higher than that trained by a decision tree.

### 3.2. Classification Results with Selected Attributes

Clearly, not all the attributes in hospitals A and B are related to the prediction of the classification. This paper used best-first and genetic algorithms to determine which attributes relate to the perdition at first, as is depicted by step 2 in Figure 1. This paper incorporates packages from Weka, version 4.1, to perform these two filtering algorithms. Weka is an open-source data mining and analysis software package written in Java, developed at the University of Waikato, New Zealand, and contains a collection of visualization tools and algorithms for data analysis and predictive modeling [34]. For hospital A, the results of the best-first algorithm are the selection of 9 attributes, including gender, age, R.B.C, MCV, N.seg., Mono, Eosi, K, and platelet, while the results of the genetic-search algorithm are the selection of 11 attributes, including gender, age, W.B.C, MCV, N.seg., Eosi, lymph, Na, K, glucose, and platelet. Similarly, for hospital B, the results of the best-first algorithm are that 5 attributes were selected, including age, hematocrit, lymph, Mono, and Eosi, while the results of the genetic-search algorithm are the selection of 10 attributes, including age, hemoglobin, hematocrit, MCH, N.seg., lymph, Mono, Eosi, glucose, and glutamic oxaloacetic transaminase. Next, we performed the experiments using the C4.5 classifier for hospital A with selected attributes that were filtered using best-first and genetic search algorithms, then we compared their accuracies with that of the C4.5 classifier with all attributes. For hospital A, the accuracies of the two C4.5 classifiers with the selected attributes of genetic search (its accuracy value being 96.71%) and best-first (its accuracy value being 97.06%) algorithms are both higher than the accuracy of the C4.5 classifiers with all attributes (its accuracy value being 95.42%). From the above experiment results, it can be seen that the C4.5 classifier with the best-first algorithm outperforms the C4.5 classifier with the genetic search one in terms of accuracy. Table 7 shows the confusion matrix of the accuracy (i.e., 97.06%) of the classification with the selected attributes using the best-first algorithm, then the C4.5 classifier only; it shows that the TP rates from DRG 164 to DRG 167 are 100.0%, 100.0%, 100.0%, and 88.3%, respectively.

We also performed experiments for hospital B with the C4.5 classifier with the selected attributes, which were filtered using best-first and genetic search algorithms; we then compared their accuracies with those of the C4.5 classifier but with all attributes. Additionally, similar experiments were performed using the BP classifier. Similarly, for hospital B, the accuracies of the two classifiers with the selected attributes by genetic search (its accuracy value being 93.80%) and best-first (its accuracy value being 91.21%) algorithms are both higher than the accuracy of the classification with all attributes (its accuracy value being 88.11%). However, from the results of the above experiments, it is shown that the C4.5 classifier with the best-first algorithm does not outperform the C4.5 classifier with the genetic search algorithm in terms of accuracy. Table 8 shows the confusion matrix of the accuracy (i.e., 93.80%) of the classification with the selected attributes using the generic search algorithm and then the C4.5 classifier. It also shows that the TP rates from DRG 164 to DRG 167 are 100.0%, 100.0%, 100.0%, and 75.8%, respectively.

Next, we performed another set of experiments to check whether the best-first and generic algorithms can enhance the accuracy of the BPN algorithm. In these experiments, learning rates were set from 0.1 to 1.0 with a uniformly increased interval of 0.1 under the values of momentum factors from 0.1 to 0.9, respectively. Since the combination of the experiments is too big, only the cases with the highest accuracy are shown. Table 9 summarizes the best cases with the selected attributes filtered by best-first and generic algorithms and then the BPN classifier for the data of hospitals A and B. It can be seen that the accuracies of the classification using the genetic search algorithm and the BPN classifier for hospitals A and B are both higher than those achieved by the best-first algorithm and then the BPN classifier.

The results in Table 9 reveal that using the genetic search algorithm and the BPN classifier is a better way of enhancing the performance of the classification. Note that the highest accuracies of classification with the selected attributes using genetic search and best-first algorithms and the BPN classifier (i.e., 97.49% and 94.12, as seen in Table 9) for hospital A are both lower than the accuracy of the classification with all attributes by the BPN classifier (i.e., 98.70%, seen in Section 3.1 that occurs when the learning rates are set as 0.7 and 0.8 under the values of their momentum factors being 0.7 and 0.9, respectively). Additionally, note that the highest accuracies of classification with the selected attributes using genetic search and best-first algorithms and the BPN classifier (i.e., 85.27% and 94.57, as seen in Table 9) for hospital B are also both lower than the accuracy of the classification with all attributes using the BPN classifier (i.e., 96.12%, as seen in Section 3.1, that occurs when the learning rates are set as 0.2 and 0.8 under the values of their momentum factors of 0.4 and 0.7, respectively). It is estimated that the BPN classifier with all attributes retains all information so as that the BPN can modify the weights continuously to obtain higher accuracy. Based on the results of the experiments, this paper further introduced the bagging method to the two classifiers to enhance the accuracy.

### 3.3. Classification Results with Bagging Method

Bagging is a machine learning technique [35] which allows many weak classifiers to combine their efforts, which renders a single strong classifier. In most cases, bagging can produce a better classifier to enhance accuracy. This paper incorporates the bagging method, also from the Weka package, to the C4.5 and BPN classifier to increase their accuracies, as is shown in step 3 in Figure 1. In the previous set of experiments with the data of hospital A, the highest accuracy is 97.06%, which occurs when using the C4.5 classifier associated with the best-first algorithm. Table 10 shows the confusion matrix of the C4.5 classifier incorporated using the bagging method; the table shows that the accuracy of the C4.5 classifier using the bagging method reaches a value of 97.84%, which consists of TP rates from DRG 164 to DRG 167 of 100.0%, 100.0%, 100.0%, and 91.4%, respectively. As has been shown, the incorporation of the bagging method can slightly increase accuracy.

Reviewing the experiments which used the data of hospital A when using the BPN classifier instead of C4.5, the highest accuracy obtained was 98.70%, which occurred in two cases when the learning rates were set as 0.7 and 0.8 under the values of their momentum factors as 0.7 and 0.9, respectively, and all the attributes were selected. For the two cases, the accuracies of the BPN classifier when incorporating the bagging method were 98.18% and 98.44%, respectively. It can be seen that these accuracies of the BPN classifier when incorporating the bagging method are almost the same as that of the sole BPN classifier. It can be guessed that all the variants of the BPN classifiers have similar accuracy.

As for hospital B, the results mentioned previously show that the highest accuracy is 93.80% when the C4.5 classifier is used in association with the genetic-search algorithm. Table 11 is the confusion matrix of the C4.5 classifier incorporating the bagging method. Its accuracy can reach 95.61%, which consists of TP rates from DRG 164 to DRG 167 of 100.0%, 100.0%, 100.0%, and 82.8%, respectively. As has been demonstrated, the incorporation of the bagging method can increase the accuracy of the C4.5 classifier considerably.

For the experiments which used the data of hospital B when using the BPN classifier instead of C4.5, the highest accuracy was 96.12%, which occurred in the two cases when the learning rates were set as 0.2 and 0.8 under the values of their momentum factors being 0.4 and 0.7, respectively, and when all the attributes were selected. For the two cases, the accuracies of the BPN classifier, if incorporating the bagging algorithm, become the values of 94.83% and 95.09%, respectively. It can be seen that these accuracies of the BPN classifier, if incorporating the bagging method in the BPN algorithm, are slightly lower than that of the sole BPN algorithm. However, the difference in the accuracies is within the estimated errors.

In summary of all the experiments and discussions which have occurred in these subsections, the highest accuracies under all of the combinations, namely, all attributes/selected attributes by best-first or genetic search algorithms, and the incorporation of the bagging method, are collated in Figure 5. It can be seen that the accuracies of the C4.5 classifier with the selected attributes using the best-first or genetic search algorithms are both higher than that of the C4.5 classifier with all attributes. In addition, the accuracies of the C4.5 classifier, when further incorporating the bagging method, can reach the highest accuracy among all the combinations. The results can be explained that not all attributes have an influence on DRG classification if using the C4.5 decision-tree classifier. Another reason is that the C4.5 classifier is constructed using a binary tree. The selected attributes can reduce the tree height of the classifier and increase its accuracy.

For the neuron-net-based classifier, it can be seen that the accuracies of the BPN classifier with the selected attributes or an incorporated bagging method are all lower than the BPN classifier solely. It is estimated that the BPN classifier with all attributes stores all information, meaning the neuron network can modify the weights continuously in order to obtain the highest accuracy.

We conclude that the values of the accuracy of the DRG code classification for appendectomy patients in two hospitals can reach as high as 97.84% and 98.70% under all of the possible combinations. In addition, the accuracy of the classifiers using the BPN classifier is higher than that using the C4.5 classifier when all the attributes are considered. In contrast, when selecting the related attributes, the accuracy of the C4.5 classifier then increases since the related attributes are more representative. The accuracy of the C4.5 classifier can be further improved when the bagging method is incorporated, while the accuracy of the BPN classifier when incorporating the bagging method does not increase as much as the C4.5 classifier.

## 4. Conclusions and Research Limitations

The goal of utilizing two algorithms (C4.5 and BPN) to construct the classifiers for the prediction of the DRG code for appendectomy patients using their medical data before their discharge from hospital is feasible, and the accuracy is very high. Thus, once patients with appendectomies are admitted, the pre-determination of the DRG codes can be used to control and decrease medical expenses and resources and improve the quality of the medical care they receive. The paper is a practical one, though it did not propose a novel technique. We hope that the results will influence the clinical guidelines and protocols in terms of caring for appendectomy patients. With the earlier classification, hospitals which are in competition with each other would only be able to effectively allocate their healthcare resources to the desired treatments in order to achieve one of the sustainable development goals. This paper details the performance (accuracy, precision, and recall) of the classification techniques applied to appendectomy patients, and the classification techniques used here can be further embedded in a clinical decision system in order to facilitate the making of informed decisions regarding the arrangement of staffing levels, equipment needs, and other resource allocations.

The filtered attribute achieved by best-first and generic search algorithms can contribute to the accuracy of the C4.5 classifier, but the increase in the accuracy of the BPN classifier is not as obvious. However, a hospital should record the items used in medical examinations that are chosen as representative attributes in time in order to predict the DRGs of the inpatients as soon as they are admitted. If the computing power allows and the time allocated to wait for the prediction result is long enough, the accuracy of the classifiers could be further enhanced by incorporating the bagging method.

This paper demonstrates that the C4.5 and BPN classifiers and their variants have high accuracy in classifying patients with appendectomies into the appropriate DRG codes. Similar processes can be applied to other DRG diseases in order to obtain a potential reimbursement before the patient is discharged from hospital. As medical records in different hospitals may have different record fields, the classifier, and the results cannot be directly applied to all hospitals. In other words, a specific DRG classification model should be constructed individually based on the types and characteristics of the medical records kept in the hospital. The classifier in this paper can be further enhanced, for example, by setting the weights of examination records and filtering the unrelated attributes in order to obtain more training data.

## Figures and Tables

**Figure 1 healthcare-11-01598-f001:**
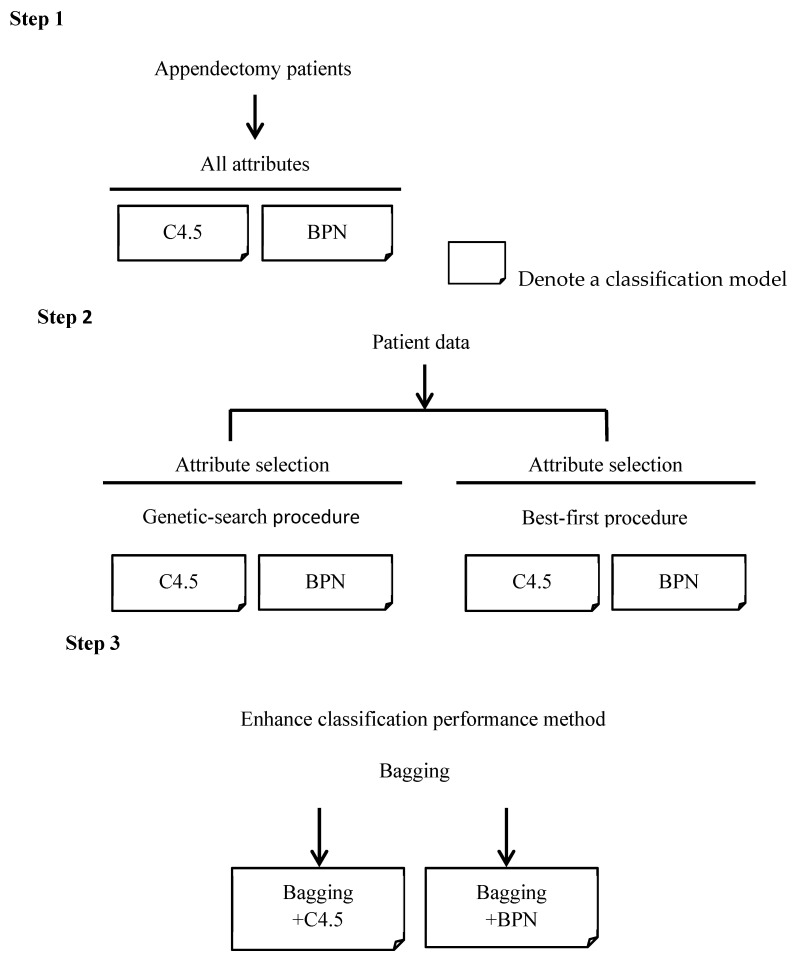
Steps of classifier construction in this paper.

**Figure 2 healthcare-11-01598-f002:**
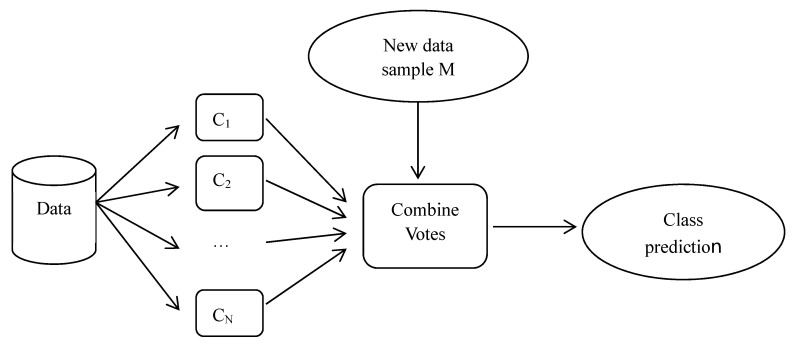
Bagging method used in this paper.

**Figure 3 healthcare-11-01598-f003:**
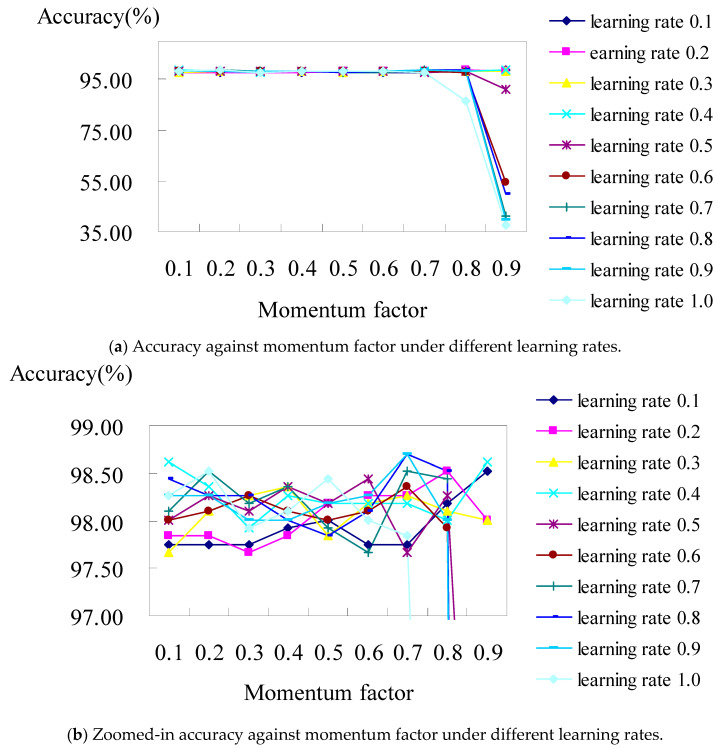
Accuracy of the BPN classifier with all attributes in hospital A.

**Figure 4 healthcare-11-01598-f004:**
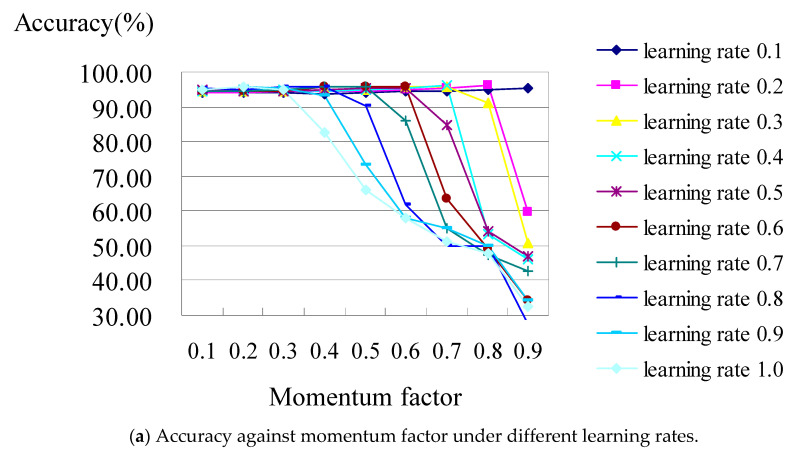
Performance of the BPN model for hospital B.

**Figure 5 healthcare-11-01598-f005:**
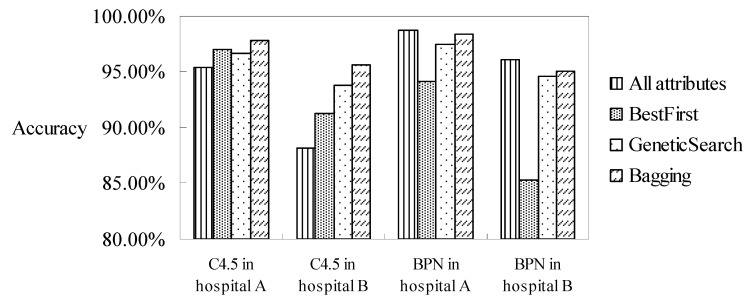
Accuracies of the classifications under the combinations of methods.

**Table 1 healthcare-11-01598-t001:** DRGs for patients with appendectomy defined by NHI, Taiwan.

DRG	Item		Principal Diagnosis **	Operating Room Procedures ***
164	Appendectomy with complicated principal diagnosis	With CC *	540.0 540.1	47.01 47.09 47.2 47.99
165	Appendectomy with complicated principal diagnosis	Without CC		
166	Appendectomy without complicated principal diagnosis	With CC	540.9 541 542 543.0 543.9	47.01 47.09 47.2 47.99
167	Appendectomy without complicated principal diagnosis	Without CC		

Note: * CC—complication and comorbidity. ** 540.0—acute appendicitis with generalized peritonitis. 540.1—acute appendicitis with peritoneal abscess. 540.9—acute appendicitis without mention of peritonitis. 541—appendicitis but unqualified. 542—other appendicitis. 543.0—hyperplasia of appendix (lymphoid). 543.9—other and unspecified diseases of appendix. *** 47.01—laparoscopic appendectomy. 47.09—other appendectomy. 47.2—drainage of appendicle abscess. 47.99—other operations on appendix.

**Table 2 healthcare-11-01598-t002:** The original and augmented sampling distribution of DRGs of patients with appendectomy in hospitals A and B.

	Hospital A			Hospital B		
	Original Samples	Ratio	Copying Samples	Original Samples	Ratio	Copying Samples
DRG 164	1	290 times	290	2	49 times	98
DRG 165	6	48 times	288	19	5 times	95
DRG 166	16	18 times	288	5	19 times	95
DRG 167	290	1 time	290	99	1 time	99
Total	313		1156	125		387

**Table 3 healthcare-11-01598-t003:** Vital sign data recorded for patients with appendectomy in hospitals A and B.

Predicting Attribute	A	B	Predicting Attribute	A	B	Predicting Attribute	A	B
Gender	v	v	MCHC	v	v	Creatinine	v	v
Age	v	v	Na (sodium)	v	v	Glutamic oxaloacetic transaminase	v	v
Hemoglobin	v	v	K (potassium)	v	v	Blood urea nitrogen	v	v
Hematocrit	v	v	Glucose	v	v	Glutamic pyruvic transaminase	v	
N. seg.	v	v	Platelet	v		Bleeding time	v	
Lymph	v	v	Occult blood		v	Clotting time	v	
Mono	v	v	specific gravity		v	Amylase	v	
Eosi	v	v	PH (urine)		v	Nephelometry (chlamydia trachomatis Ag)	v	
Baso	v	v	Urobilinogen		v	RDW-CV		v
W.B.C	v	v	Color (urine)		v	Mean platelet volume		v
R.B.C	v	v	Clarity (urine)		v	Prothrombin time		v
MCV	v	v	R.B.C (urine)		v	Activated partial thromboplastin time		v
MCH	v	v	W.B.C (urine)		v	Ep. Cell		v

Note: ‘v’ denotes prediction attributes that appear in hospitals A or B.

**Table 4 healthcare-11-01598-t004:** Confusion matrix of the C4.5 classifier with all attributes.

**(a) Hospital A with 26 Prediction Attributes**						
**Number of Patients’ Records**		**Predicted Class**				
		**DRG 164**	**DRG 165**	**DRG 166**	**DRG 167**	**Total**
Actual class	DRG 164	290	0	0	0	290
	DRG 165	0	288	0	0	288
	DRG 166	0	0	288	0	288
	DRG 167	2	25	26	237	290
	Total	292	313	314	237	1156
**(b) Hospital B with 33 Prediction Attributes**						
**Number of Patients’ Records**		**Predicted Class**				
		**DRG 164**	**DRG 165**	**DRG 166**	**DRG 167**	**Total**
Actual class	DRG 164	98	0	0	0	98
	DRG 165	0	87	2	6	95
	DRG 166	0	0	95	0	95
	DRG 167	2	25	11	61	99
	Total	100	112	108	67	387

**Table 5 healthcare-11-01598-t005:** Confusion matrix of the BPN classifier with all attributes in hospital A.

**(a) The learning rate is 0.8 under the value of a momentum factor of 0.7.**						
**Number of Patients’ Records**		**Predicted Class**				
		**DRG 164**	**DRG 165**	**DRG 166**	**DRG 167**	**Total**
Actual class	DRG 164	290	0	0	0	290
	DRG 165	0	288	0	0	288
	DRG 166	0	0	288	0	288
	DRG 167	2	4	9	275	290
	Total	292	292	297	275	1156
**(b) The Learning Rate is 0.9 under the Value of the Momentum Factor Being 0.7.**						
**Number of Patients’ Records**		**Predicted Class**				
		**DRG 164**	**DRG 165**	**DRG 166**	**DRG 167**	**Total**
Actual class	DRG 164	290	0	0	0	290
	DRG 165	0	288	0	0	288
	DRG 166	0	0	288	0	288
	DRG 167	1	4	10	275	290
	Total	291	292	298	275	1156

**Table 6 healthcare-11-01598-t006:** Confusion matrix of the BPN classifier with all attributes in hospital B.

**(a) The Learning Rate is 0.2 under the Value of a Momentum Factor of 0.8.**							
**Number of Patients’ Records**			**Predicted Class**				
			**DRG 164**	**DRG 165**	**DRG 166**	**DRG 167**	**Total**
Actual class	DRG 164		98	0	0	0	98
	DRG 165		0	95	0	0	95
	DRG 166		0	0	95	0	95
	DRG 167		2	12	1	84	99
	Total		100	107	96	84	387
**(b) The Learning Rate is 0.4 under the Value of a Momentum Factor Being 0.7.**							
**Number of Patients’ Records**			**Predicted Class**				
			**DRG 164**	**DRG 165**	**DRG 166**	**DRG 167**	**Total**
Actual class		DRG 164	98	0	0	0	98
		DRG 165	0	95	0	0	95
		DRG 166	0	0	95	0	95
		DRG 167	2	13	0	84	99
		Total	100	108	95	84	387

**Table 7 healthcare-11-01598-t007:** Confusion matrix with selected attributes by best-first and C4.5 classifier for hospital A.

Number of Patients’ Records		Predicted Class				
		DRG 164	DRG 165	DRG 166	DRG 167	Total
Actual class	DRG 164	290	0	0	0	290
	DRG 165	0	288	0	0	288
	DRG 166	0	0	288	0	288
	DRG 167	2	14	18	256	290
	Total	292	302	306	256	1156

**Table 8 healthcare-11-01598-t008:** Confusion matrix with selected attributes by generic search and C4.5 classifier for hospital B.

Number of Patients’ Records		Predicted Class				
		DRG 164	DRG 165	DRG 166	DRG 167	Total
Actual class	DRG 164	98	0	0	0	98
	DRG 165	0	95	0	0	95
	DRG 166	0	0	95	0	95
	DRG 167	1	21	2	75	99
	Total	99	116	97	75	387

**Table 9 healthcare-11-01598-t009:** Cases of highest accuracy with the selected attributes by best-first and generic search algorithm and BPN classifier.

Accuracy (%)	Selection Attributes Method	
	Best-First	Genetic Search
Hospital A	Learning rate 0.1 Momentum factor 0.1 Accuracy 94.12%	Learning rate 0.5 Momentum factor 0.3 Accuracy 97.49%
		Learning rate 0.6 Momentum factor 0.4 Accuracy 97.49%
Hospital B	Learning rate 0.9 Momentum factor 0.1 Accuracy 85.27%	Learning rate 0.7 Momentum factor 0.6 Accuracy 94.57%

**Table 10 healthcare-11-01598-t010:** Confusion matrix of the C4.5 classifier with bagging method for hospital A.

Number of Patients’ Records		Predicted Class				
		DRG 164	DRG 165	DRG 166	DRG 167	Total
Actual class	DRG 164	290	0	0	0	290
	DRG 165	0	288	0	0	288
	DRG 166	0	0	288	0	288
	DRG 167	2	11	12	265	290
	Total	292	299	300	265	1156

**Table 11 healthcare-11-01598-t011:** Confusion matrix of the C4.5 classifier with bagging method for hospital B.

Number of Patients’ Records		Predicted Class				
		DRG 164	DRG 165	DRG 166	DRG 167	Total
Actual class	DRG 164	98	0	0	0	98
	DRG 165	0	95	0	0	95
	DRG 166	0	0	95	0	95
	DRG 167	1	15	1	82	99
	Total	99	110	96	82	387

## Data Availability

Not applicable.

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
