# Peer review of "Prediction of Diagnosis-Related Groups for Appendectomy Patients Using C4.5 and Neural Network"

_healthcare, 2023, doi:10.3390/healthcare11111598_

Round 1
Reviewer 1 Report (Previous Reviewer 1)
I appreciate the clarifications provided by the authors in the revised manuscript. I have no specific remarks on the conduction and presentation of the research just a minor issue: revise the syntax in the sentence in lines 167-168.
Author Response
Thanks for the indication of typos in lines 167-168. We have modified the sentences in the specified lines and ask the English editing service to screen and smooth the paper.
Reviewer 2 Report (Previous Reviewer 4)
Revised version can be accepted.
Author Response
We appreciated the confirmation of our efforts. Thanks a lot.
Reviewer 3 Report (New Reviewer)
The paper is scientifically interesting, and the rationale for the need and purpose of the study given in the introduction is sufficiently convincing.
The authors should think about providing all necessary information needed for others to reproduce their results. At minimum, in the methods section it would be a good idea to mention all the software packages used as well as the corresponding version numbers, and to cite them.
A serious concern is that the writing is very poor because English is not the author's first language. I know that MDPI provides a language editing service that is very helpful, but my worry is that some of the language editing may need to be done by somebody that understands the scientific content of the paper. It would be better if the authors had assistance from a proficient English speaker in this area of research.
Page 8, lines 260-268: The definition of precision and recall rates is very important. The verbal definition is not well written and very confusing and needs to be edited by someone that understands the definition. The authors could also clarify the definition by explaining how it is calculated using the numbers available in the confusion matrix. For example, to calculate the precision rate, one is adding the diagonal elements on the confusion matrix and then divides by the lower right corner Total number. For the recall rates, there is one for each diagnostic code, and they are calculated by dividing the corresponding diagonal element on the confusion matrix, with the element on the total column, and corresponding row. This becomes clear later from the example on page 9 lines 288-295. The authors should also make the terminology consistent since what is initially defined as the precision rate is subsequently referred to as the accuracy rate.
The main concern that I have is improving the language, because with the current version of the manuscript, it is too distracting, and not acceptable for publication. However, I don't see any reason why it couldn't be corrected.
Round 2
Reviewer 3 Report (New Reviewer)
The authors made substantial revisions to the manuscript that corrected the concerns that I had about the language. They have also addressed the other concerns that I raised about the definitions of precision rate, accuracy rate, and recall rate, and they have mentioned and cited the software used for their calculations. Based on the above, I recommend that the paper be accepted.
This manuscript is a resubmission of an earlier submission. The following is a list of the peer review reports and author responses from that submission.
Round 1
Reviewer 1 Report
In today's healthcare landscape, where costs are rising and resources are limited, this study aims to predict the Diagnosis Related Group (DRG) of inpatients who have undergone appendectomy as early as possible.
I see a major issue in the proposed manuscript.
The question arises as to why knowing the reimbursement of inpatients beforehand, and hence predicting a patient's DRG "as soon as possible," perhaps even at admission, is important for efficient resource allocation. It may not be immediately apparent how a model developed to forecast economic outcomes, such as the final DRG, could impact the diagnosis and treatment path of an individual patient, which is grounded in guidelines, local protocols, and refined on a case-by-case basis by physicians.
However, it is possible that physicians could benefit from a clinical decision support system that provides them with insights and suggestions. This, in the long term, could also be used by hospital administrators and policy makers to guide resource allocation decisions. In fact, by having a better understanding of the likely DRGs of incoming patients, administrators can make more informed decisions about staffing levels, equipment needs, and other resource allocation decisions.
In conclusion, while predicting a patient's DRG may not directly impact their diagnosis and treatment path, it has the potential to inform more efficient resource allocation decisions and ultimately lead to better patient outcomes.
Other minor comments and suggestions for the introduction are reported directly in the manuscript.
Best regards

Reviewer 2 Report
In the paper it is not clear which are the novelties presented here compared to the existing literatures. In my opinion the paper does not add enough new facts to be sufficiently interesting for publication in Healthcare.
However, I see no new ideas here.
In addition, the abstract of this paper needs to be rewritten. The first half of the summary needs a lot of deletion and simplification.
Reviewer 3 Report
The authors must to review totally the paper:
The methods how the data was collected, if good practices were applied, in the collection of the dataset, if no bias is present, etc.
Data augmentation is used, however, a comparison with the minority classess is required to ensure that the oversampling method is not introducing further bias.
Other relevant metrics should be added to better assess the performance of the paper. Accuracy shall be complemented with recall, and the rates of TPs, FPs are mentioned, but not sufficiently explained.
Results shown in the tables are not sufficiently explained. It is hard to understand why is there an accuracy of 100& in some groups, whereas there is not a justified reason for the dispersion for DRG 167 in all tables.
In general, the paper is hard to read and follow, it requires a complete review. There are multiple typos and sentences with no sense.
The conclusions are rather poor. The use of genetic algorithms for variable selection is mentioned, but the impact on the performance of the NNs is not described.
Reviewer 4 Report
My comments are given below:
1. Mention in short, the main results, conclusion in Abstract.
2. Compare the results with the article [https://academic.oup.com/bjs/article/107/1/73/6120962], mention the main motivation of this research.
3. Also mention from where the data of this research were collected.
4. Check the typos and grammatical errors of the paper. Reference are not in unique formatting.
5. In Figure 4(b), the color for "earning rate 1.0"is not clearly identified. If possible use another color.
6. Comment on how the results obtained here will be useful for health organization.